# Prolonged oscillating preoptic area kisspeptin neuron activity underlies the preovulatory luteinizing hormone surge in mice

**Ziyue Zhou, Cheng-Yu Huang, Allan Edward Herbison***

Department of Physiology, Development and Neuroscience, University of Cambridge, Cambridge, United Kingdom

## eLife Assessment

This **fundamental** work advances our understanding of the role of kisspeptin neurons in regulating the luteinizing hormone (LH) surge in females. The study uses cutting-edge techniques to provide **compelling** and rigorous data supporting a critical role of RP3V kisspeptin neurons in the neuroendocrine LH surge process. This research will be of interest to reproductive biologists and neuroscientists studying the female ovarian cycle. Continuing to examine the complexities of the LH surge and the neuronal populations involved, as done in this study, is critical for developing therapeutic treatments for women's reproductive disorders.

**\*For correspondence:**
aeh36@cam.ac.uk

**Competing interest:** The authors declare that no competing interests exist.

**Abstract** The population of kisspeptin neurons located in the rostral periventricular area of the third ventricle (RP3V) is thought to have a key role in generating the GnRH surge that triggers ovulation. Using a modified GCaMP fibre photometry procedure, we have been able to record the in vivo population activity of RP3V$^{KISS}$ neurons across the estrous cycle of female mice. A marked increase in GCaMP activity was detected beginning on the afternoon of proestrus that lasted in total for 13±1 hr. This was comprised of slow baseline oscillations with a period of 91±4 min associated with high-frequency rapid transients. Very little oscillating baseline or transient activity was detected at other stages of the estrous cycle. Concurrent blood sampling showed that the peak of the LH surge occurred 3.5±1.1 hr after the first baseline RP3V$^{KISS}$ neuron baseline oscillation on the afternoon of proestrus. The time of onset of RP3V$^{KISS}$ neuron oscillations varied between mice and across subsequent proestrous stages in the same mice. To assess the impact of estradiol on RP3V$^{KISS}$ neuron activity, mice were ovariectomized and given an incremental estradiol replacement regimen. Minimal patterned GCaMP activity was found in OVX mice, and this was not changed acutely by any of the estradiol treatments. However, on the afternoon of the expected LH surge, the same oscillating baseline activity with associated transients occurred for 7.1±0.5 hr. These observations reveal an unexpected prolonged oscillatory pattern of RP3V$^{KISS}$ neuron activity that is dependent on estrogen and underlies the preovulatory LH surge as well as potentially other facets of reproductive behavior.

## Introduction

A neural surge generator is responsible for integrating multiple signals to control the mid-cycle activation of GnRH neurons that triggers the preovulatory luteinizing hormone (LH) surge (*Herbison, 2015*; *Kauffman, 2022*). It now seems clear that a preoptic area population of kisspeptin neurons

that directly activate GnRH neuron cell bodies (*Piet et al., 2018*) is an essential component of the surge generator in spontaneously ovulating mammals (*Matsuda et al., 2019*; *Goodman et al., 2022*).

The primary regulator of surge generator activity in spontaneously ovulating species is the gradually increasing levels of circulating estradiol that occur throughout the follicular phase of the cycle. In rodents, this '"estrogen positive feedback' mechanism is mediated by estrogen receptor alpha (ESR1) and occurs within the rostral periventricular area of the third ventricle (RP3V) (*Wintermantel et al., 2006*; *Porteous and Herbison, 2019*); a preoptic brain region encompassing the anteroventral periventricular nucleus (AVPV) and periventricular preoptic nucleus (PeN) (*Herbison, 2008*). In vivo CRISPR/Cas9 gene editing to knockdown ESR1 expression selectively in RP3V kisspeptin (RP3V[KISS]) neurons has been shown to suppress both the LH surge and estrous cyclicity in adult female mice (*Wang et al., 2019*; *Clarkson et al., 2023*). Although of variable importance amongst mammalian species, a circadian input to the surge generator is also critical for the GnRH surge in rodents (*Goodman et al., 2022*). It remains unclear how this operates, but a vasopressin input from the suprachiasmatic nucleus to the RP3V[KISS] neurons is suspected to be important in triggering the onset of the surge (*Tonsfeldt et al., 2022*; *Piet, 2023*).

The ability to monitor the activity of arcuate nucleus (ARN) kisspeptin neurons in freely behaving mice with GCaMP fibre photometry has been invaluable in defining and then understanding their role as the GnRH pulse generator in puberty and adulthood, as well as in pathological states (*Clarkson et al., 2017*; *Han et al., 2019*; *McQuillan et al., 2019*; *Liu et al., 2021*; *McQuillan et al., 2022*; *Goto et al., 2023*; *Han et al., 2023*; *Goto et al., 2025*; *Zhou et al., 2025*). To date, attempts to record the behavior of the RP3V[KISS] neurons have been unsuccessful due primarily to the vertical column-like topography of the RP3V[KISS] neurons alongside the third ventricle. Indeed, attempts by us to employ bevelled or mirror lenses that gather fluorescence from the side of the optic fibre have been met with very limited success. We now report here the use of tapered optic fibres (*Pisano et al., 2019*) that enable the population activity of RP3V[KISS] neurons to be monitored in real time in freely behaving mice. This reveals the estrogen-dependent dynamics of RP3V[KISS] neuron population activity across the estrous cycle and demonstrates an unexpected oscillatory pattern of synchronized activity that continues for over 12 hr on proestrus.

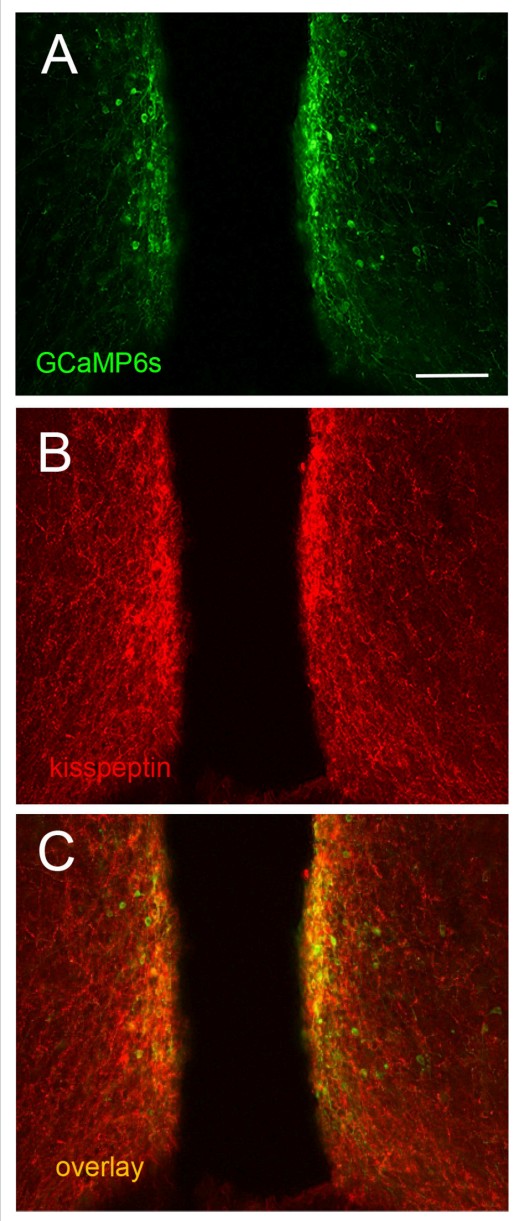

**Figure 1.** GCaMP expression in rostral periventricular area of the third ventricle kisspeptin (RP3V[KISS]) neurons. Confocal images of (**A**) GFP (GCaMP6s, green) and (**B**) kisspeptin (red) immunofluorescence in the RP3V, and (**C**) an overlay of both in the periventricular nucleus of a proestrous female mouse. Scale bar = 100 μm.

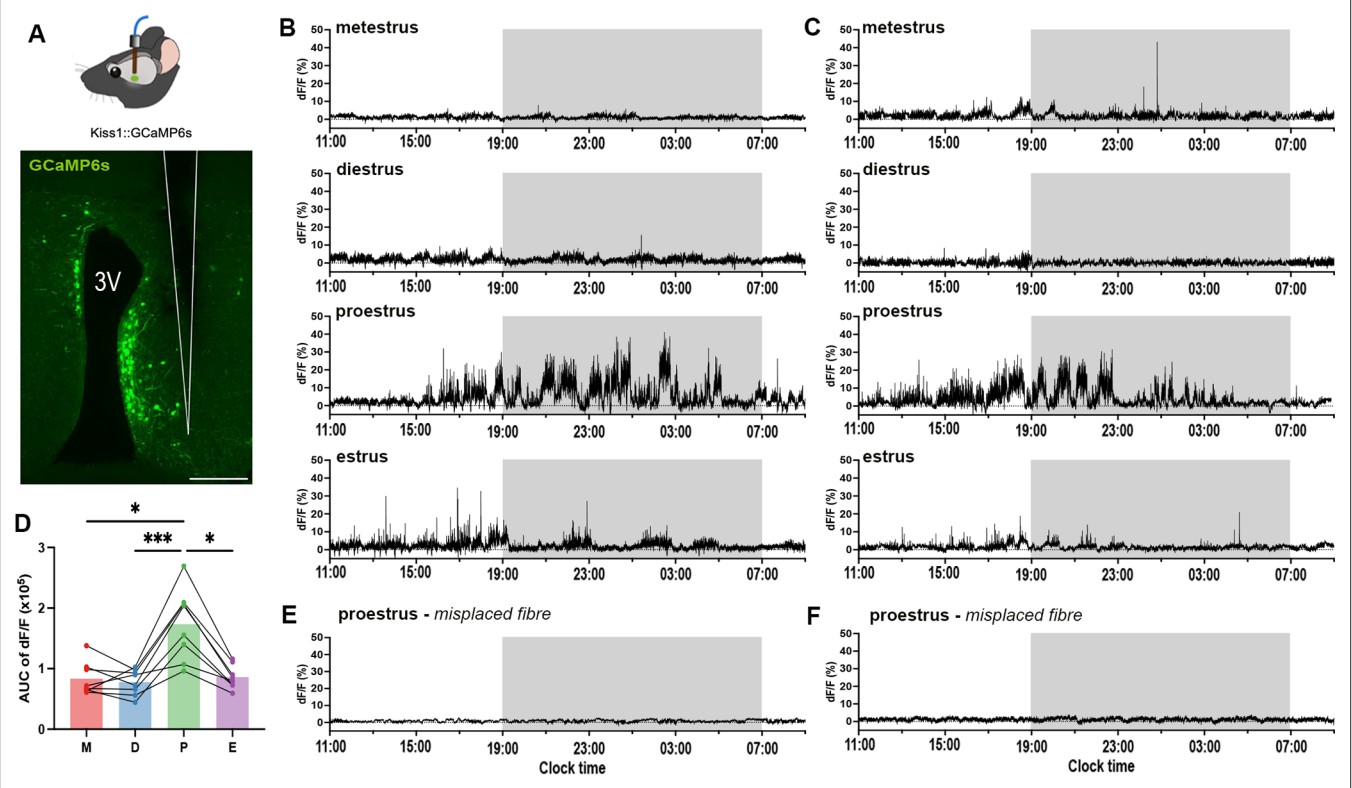

**Figure 2.** GCaMP signals in rostral periventricular area of the third ventricle kisspeptin (RP3V^KISS) neurons across the estrous cycle. (**A**) Coronal section showing location of tapered optic fibre (white outline) in relation to GCaMP-expressing cells (green) lining the third ventricle (3 V). Scale bar = 100 μm. (**B** and **C**) Two representative examples of 22 hr GCaMP fibre photometry recordings across the complete estrous cycle in two mice. Light-off period is represented by gray shaded area. (**D**) Mean area under the curve (AUC) calculated from 22 hr recordings across the estrous cycle: metestrus (M), diestrus (D), proestrus (P), and estrus (E). Each dot represents an animal (N=8). Friedman test followed by Dunn's multiple comparisons tests. *$p<0.05$, ***$p<0.001$. (**E** and **F**) Examples of 22 hr GCaMP fibre photometry recordings from two proestrous female mice with misplaced optic fibers where signals remain <3% of baseline.

## Results

### Characterization of GCaMP expression in RP3V^KISS neurons

Adult *Kiss1^{Cre/+}* (Palmiter v2) female mice (*Padilla et al., 2018*) were injected bilaterally with recombinant Cre-dependent AAVs encoding GCaMP6s. As the relationship of Cre-driven expression to kisspeptin immunoreactivity has not been reported for the RP3V in this line, we undertook dual label immunofluorescence for GFP (GCaMP6s) and kisspeptin in five female mice killed on proestrus. All mice exhibited many GFP-immunoreactive cells (33.0±5.9 cells/section) located primarily adjacent to the third ventricle within the preoptic area (*Figure 1A*) and this exhibited overlap with kisspeptin immunoreactivity (21.8±3.1 cells/section) (*Figure 1B and C*). However, the number of clearly identifiable dual labeled cells were 10.4±2.5/section resulting in 47.0±5.7% of kisspeptin-immunoreactive neurons expressing GCaMP6s and 33.0±5.5% of GFP cells expressing clearly cytoplasmic kisspeptin immunoreactivity. These relatively low levels of Cre-targeted expression, compared to ARN kisspeptin neurons, are typical for RP3V kisspeptin neurons in other Kiss1-Cre lines (*Yip et al., 2015*; *Yeo et al., 2016*; *Piet et al., 2018*). This presumably results from the difficulty of detecting kisspeptin immunoreactivity in the cytoplasm of all RP3V kisspeptin neurons. *Kiss1-Cre* negative proestrus female mice given AAV injections (N=2) did not exhibit any GFP/GCaMP6s expression.

### RP3V^KISS neuron population activity across the estrous cycle

To make fibre photometry recordings from RP3V kisspeptin neurons, mice were given unilateral injections of AAV9-CAG.FLEX.GCaMP6s into the RP3V of adult *Kiss1^{Cre/+}* female mice followed by the implantation of a tapered optic fibre (*Figure 2A*). Three weeks later, mice were recorded continuously

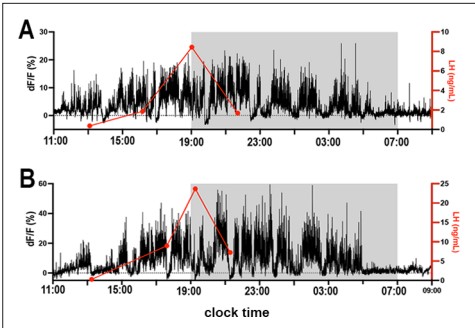

**Figure 3.** Relationship of rostral periventricular area of the third ventricle kisspeptin (RP3V^KISS) neuron activity to the LH surge on proestrus. (**A, B**) Two representative examples showing the relationship between the increase in GCaMP activity (black) with luteinizing hormone (LH) surge (red) assessed with 2–4 hr tail tip blood samples. Light-off period is represented by gray shaded area.

across each day of the estrous cycle. The GCaMP signals recorded from RP3V^KISS neurons (N=8) showed minimal fluctuations during metestrus and diestrus but became markedly more active on the afternoon of proestrus displaying an oscillatory pattern of increased baseline fluorescence associated with high frequency transients (*Figure 2B and C*). This heightened activity gradually subsided by the morning of estrus before returning to a quiescent state (Fig. B, C). An assessment of relative activity across the estrous cycle using area under the curve (AUC) demonstrated a significant increase in GCaMP fluorescence on proestrus compared to all of the other stages (Friedman Test: $\chi^2$=16.2, *p*=0.0010; Dunn's multiple comparisons tests: *p*=0.0402 (metestrus), *p*=0.006 (diestrus), *p*=0.0402 (estrus); *Figure 2D*). All five mice were found to have optic fibers running immediately alongside and lateral to GCaMP-expressing cells in the AVPV/PeN. Two mice with misplaced optic fibres exhibited almost no baseline activity (*Figure 2E and F*).

## RP3V^KISS neuron population activity in relation to the LH surge on proestrus

To examine the relationship between the increased activity of RP3V^KISS neurons and the LH surge on proestrus, tail-tip bleeding was undertaken at 2–4 hr intervals for 8–10 hr (N=5). Peak LH levels (18.0±3.2 ng/ml) occurred 3.5±0.4 hr after the first RP3V^KISS neuron oscillation was detected with oscillations then continuing for a further 7.4±1.6 hr after the decline in LH levels (*Figure 3A and B*).

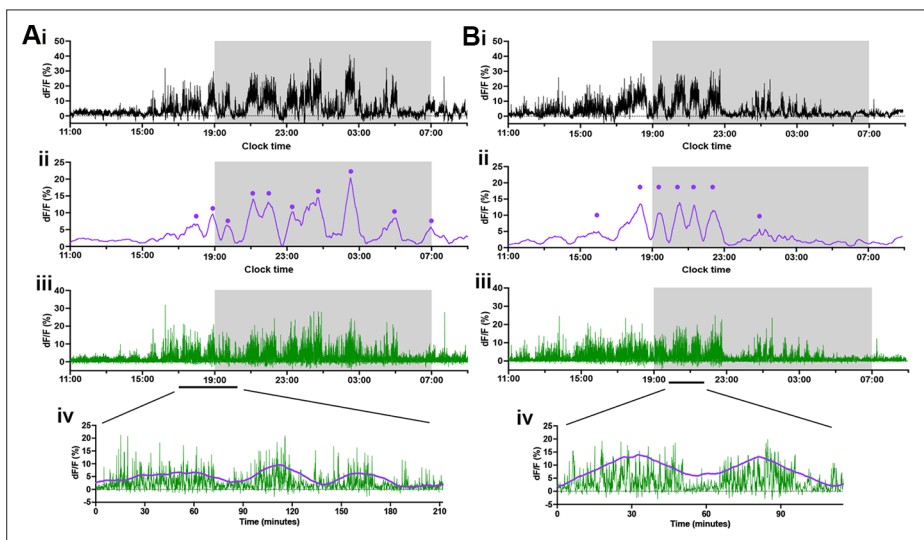

**Figure 4.** Deconvolution of rostral periventricular area of the third ventricle kisspeptin (RP3V^KISS) neuron GCaMP signals. (**A, B**) Representative examples of 22 hr photometry recordings in proestrus from the same two female mice in *Figure 2* showing (i) the original recording, (ii) a 30 min moving average highlighting the baseline oscillations (purple) with identified oscillations labeled with dots, (iii) high frequency transients (green) after the moving average is subtracted from the original recording, and (iv) expanded views of the traces showing moving average and high frequency transients. Light-off period is represented by gray shaded area.

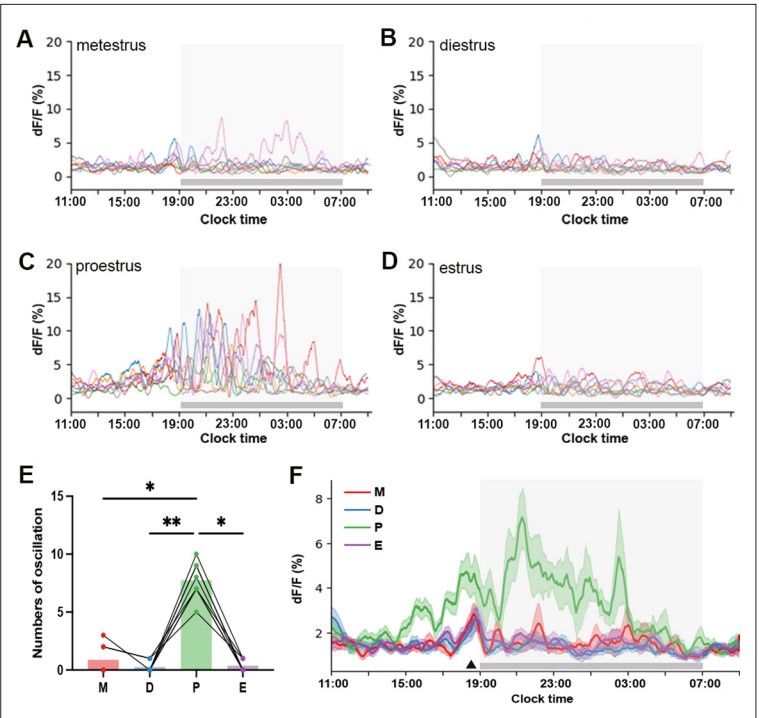

**Figure 5.** Baseline oscillations of rostral periventricular area of the third ventricle kisspeptin (RP3V^KISS) neurons during proestrus. (**A–D**) Individual baseline oscillation traces from 22 hr recordings for each animal across each of the estrous cycle stages (n=8). (**E**) Mean number of oscillations identified across each stage of the estrous cycle. Each dot represents an animal Friedman Test: $\chi^2$=18.75, $p$=0.0003; Dunn's multiple comparisons tests versus proestrus: $p$=0.0402 (metestrus), $p$=0.0029 (diestrus), $p$=0.0117 (estrus). (**F**) Mean baseline oscillations showing variations across the estrous cycle: metestrus (M, red), diestrus (D, blue), proestrus (P, green), and estrus (E, purple). Shaded regions around the traces indicate ± SEMs of corresponding colors (N=8). Triangle marks the oscillation occurring before lights-off. Light-off period is represented by gray shaded area.

## Oscillating baseline shifts in RP3V^KISS neuron population activity on proestrus

To provide a detailed analysis of the GCaMP signals recorded on proestrus, a customized MATLAB code was used to separate slow and fast signal components. A 30 min moving average was applied to the original dF/F signal (*Figure 4A i and B i*) to extract the slow baseline changes in GCaMP activity (*Figure 4A ii and B ii*). Subtraction of this baseline from the original signal leaves the high frequency transients (*Figure 4A iii and B iii*).

The baseline shifts in GCaMP fluorescence exhibited an oscillatory pattern on proestrus in all eight mice (*Figure 4A ii and B ii* and *Figure 5C*). The first identified oscillation was observed to begin 3.7±0.5 hr (range: 1.6–5.3 hr) before lights-off with the total duration of oscillatory behavior lasting 12.7±0.7 hr (range: 10.0–15.0 hr) (*Figure 5C and E*). The mean duration of each slow oscillation was 90.8±4.4 min (range: 39.3–166.1 min). Oscillations were occasionally observed in other stages of the cycle (3 of the 8 mice) and these had similar durations (metestrus: 88.6±9.5 min, diestrus: 120.4±29.7 min, estrus: 91.9±21.7 min; Kruskal-Wallis test: $\chi^2$=1.65, $p$=0.65). The numbers of oscillations observed on proestrus were substantially greater than at all other stages of the cycle (Friedman Test: $\chi^2$=18.75, $p$=0.0003; Dunn's multiple comparisons tests: $p$=0.0402 (metestrus), $p$=0.0029 (diestrus), $p$=0.0117 (estrus); *Figure 5E*) and had significantly higher amplitudes (Friedman Test: $\chi^2$=17.86, $p$=0.0005; Dunn's multiple comparisons tests: proestrus 7.2±0.9; metestrus 1.9±1.0 $p$=0.0221, diestrus 1.1±0.8 $p$=0.0084, estrus 1.5±0.8 $p$=0.0084). Interestingly, a consistent ~90 min oscillation started 1 hr preceding 'lights-off' throughout the estrous cycle (*Figure 5F*).

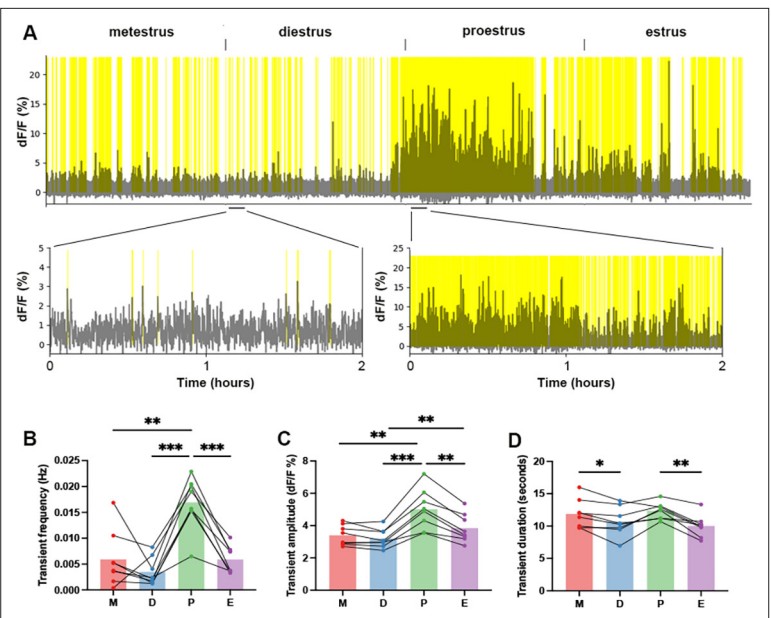

**Figure 6.** Changes in rostral periventricular area of the third ventricle kisspeptin (RP3V^KISS) neuron high frequency transients across the cycle. (**A**) An example of high frequency transient signals across the estrous cycle in one mouse. Identified significant transients are highlighted in yellow and expanded views given below. Note different y-axes for diestrus and proestrous graphs. (**B**) Mean frequency (Hz), (**C**) amplitude (dF/F, %) and (**D**) duration (seconds) of transients in metestrus (M), diestrus (D), proestrus (P), and estrus (E). Each dot represents an animal (N=8). Repeated measures one-way ANOVA followed by Tukey's multiple comparisons test; *$p<0.05$, **$p<0.01$, ***$p<0.001$ (see text for exact p values).

## High frequency transient activity in RP3V^KISS neuron population activity

After baseline correction, high frequency transients were identified using a z-scoring approach, where dF/F traces were normalized and thresholded across multiple values (k=1–4, step 0.2). This method provides a data-driven, threshold-independent way to detect transients while reducing the impact of noise and ensuring robustness across different signal amplitudes.

Transient activity was detected throughout the estrous cycle but had its highest frequency on proestrus (Repeated measures one-way ANOVA: $F=24,35$, $p<0.0001$; Tukey's multiple comparisons tests: $p=0.0025$ (metestrus), $p=0.0003$ (diestrus), $p=0.0008$ (estrus); *Figure 6A and B*). Similarly, the amplitude of individual transients was higher in proestrus compared to other stages (Repeated measures one-way ANOVA: $F=35.58$, $p=0.0001$; Tukey's multiple comparisons tests: $p=0.0032$ (metestrus), $p=0.0010$ (diestrus), $p=0.0014$ (estrus); *Figure 6A and C*). Each transient typically had a duration of ~10 s and this changed slightly across the cycle being longer in proestrus compared to estrus (Repeated measures one-way ANOVA: $F=7.71$, $p=0.0059$; Tukey's multiple comparisons tests: $p=0.0053$), with transients also being longer in metestrus when compared to diestrus (Tukey's multiple comparisons tests: $p=0.040$) (*Figure 6D*). When aligning high frequency transients with baseline oscillations, the up-state of each oscillation appears to coincide with an increased frequency of transients, whereas the troughs in the baseline correlate with a slower frequency of transients (*Figure 4A iv and B iv*).

## Shifting onset of RP3V^KISS neuron population activation across subsequent proestrous stages

The onset of GnRH neuron surge activity exhibits substantial variability between mice and also within cycles in individual mice (*Han et al., 2025*; *Yeo et al., 2025*). To examine the consistency of the surge-like behavior observed in proestrus, RP3V^KISS neuron activity was recorded across two or three proestrous stages in the same mouse and GCaMP signals were deconvolved to assess baseline shifts.

All seven mice exhibited variability in the initiation and duration of oscillatory activity occurring during proestrus across different cycles (*Figure 7A and B*). On average, the peak of the first oscillation

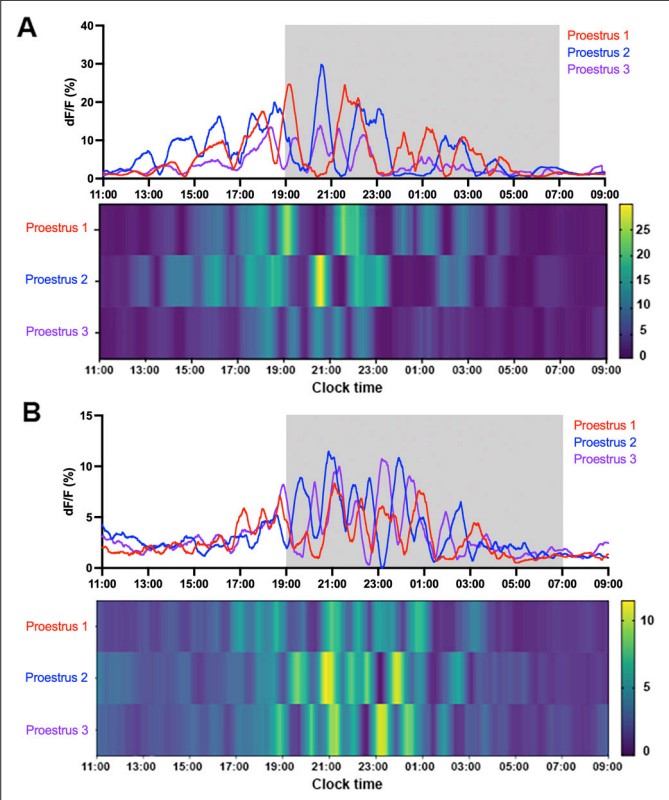

**Figure 7.** Variable oscillations in rostral periventricular area of the third ventricle kisspeptin (RP3V^KISS) neuron activity across proestrous surges in the same mice. (**A** and **B**) Two representative examples showing code-extracted baseline profiles of RP3V^KISS neuron GCaMP activity across three proestrous surges in the same mouse. Light-off period is represented by gray shaded area. Heat maps below show calcium signal dynamics (dF/F) with each row representing one proestrus. Color intensity indicates dF/F values, with yellow and green colors representing higher signal amplitudes. Legends of heat maps show color codes for dF/F (%).

varied by 2.0±0.5 hr when examining surge activity in the same mice (range: 0.4–4.2 hr; n=7). There was no consistent pattern in surge onset timing across subsequent proestrous surges with the onset fluctuating both forward and backward (*Figure 7A and B*).

## RP3V^KISS neuron population activity in OVX, estrogen-replaced female mice

Rising follicular phase levels of estradiol (E2) are the key signal driving the GnRH surge generator. To examine the effect of estradiol on RP3V^KISS neuron activity, five of the mice reported above were ovariectomized (OVX) and subjected to a well-established estrogen replacement regimen involving the placement of E2-containing Silastic capsules followed by s.c. injection of estradiol benzoate (EB) (*Bronson, 1981*; *Figure 8*). This provided within-animal comparisons between intact and OVX states, as well as the proestrous and OVX +E2+EB-induced surges.

Mice examined one week after OVX exhibited a significant decrease in RP3V^KISS neuron population activity compared with the intact (diestrus) state (*Figures 8A, B and 9A*; Paired t-test: *p*=0.002). Despite an upward trend, the subsequent replacement of E2 did not result in any significant change in RP3V^KISS neuron activity 5 days after capsule implantation (*Figures 8B, C and 9B*) (Friedman test: $\chi^2$=10.68, *p*=0.0055 for all groups; Dunn's multiple comparisons tests: *p*=0.26 (OVX vs OVX+E2)). This procedure has been shown previously to return estradiol to physiological diestrus levels (*Porteous et al., 2021*). Similarly, the injection of EB on day 6 did not have any significant effect on GCaMP signal in OVX+E2+EB mice (*Figures 8C, D and 9B*) Dunn's multiple comparisons tests: *p*>0.99 (OVX vs OVX+E2+EB). However, the following day, when an LH surge was expected, RP3V^KISS neuron activity became elevated with the appearance of baseline oscillations and enhanced transient

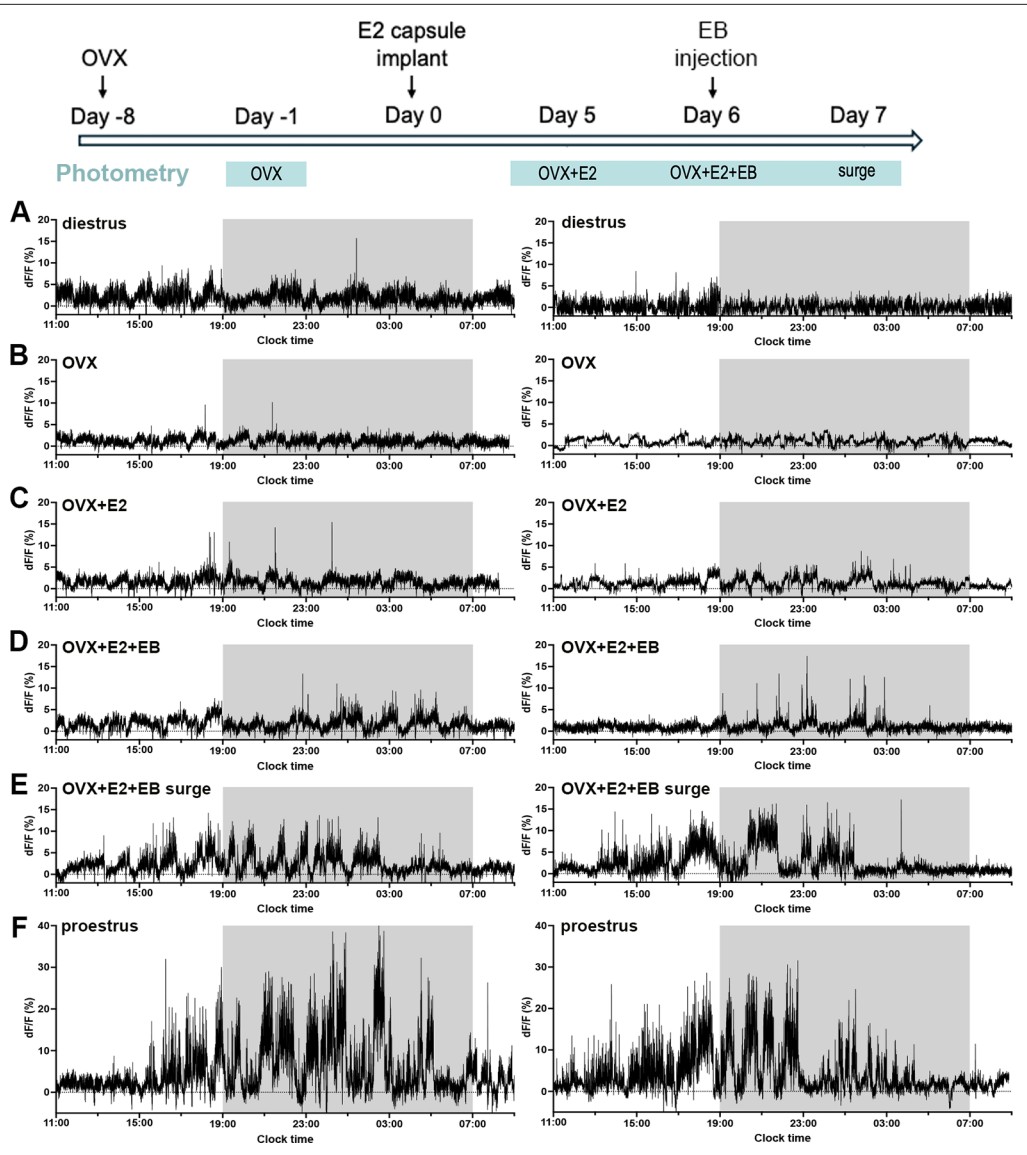

**Figure 8.** Effects of ovariectomy and estrogen replacement on rostral periventricular area of the third ventricle kisspeptin (RP3V^KISS) neuron activity. Experimental plan is shown at the top with times of photometry recordings highlighted in blue. Representative examples of 22 hr photometry recordings from two female mice (right and left) in (**A**) diestrus, (**B**) one week after ovariectomy (OVX), (**C**) five days after the estradiol implant (OVX+E2), (**D**) on the day of estradiol benzoate injection (OVX+E2+EB), (**E**) at the time of the expected estradiol-induced luteinizing hormone (LH) surge (OVX+E2+EB surge), and for comparison (**F**) in proestrus (before ovariectomy). Light-off period is represented by gray shaded area.

activity (**Figure 8E**) with a significant increase in AUC fluorescence compared with OVX mice (Dunn's multiple comparisons tests: $p=0.021$; **Figure 8B**). Qualitatively, the pattern of RP3V^KISS neuron surge activity appeared similar on proestrus and OVX +E2+EB in the same mice (**Figure 8E and F**) but the magnitude of change was substantially reduced in the OVX+E2+EB condition (paired t-test: $p=0.014$; **Figure 9C**).

To evaluate RP3V^KISS neuron activity in more detail in the OVX+E2+EB paradigm, the same deconvolution methods employed for the intact recordings were used (**Figure 9D and E**). Three of five females exhibited identified oscillations in the OVX+E2+EB surge state with 3.7±0.3 oscillations (range: 3–4 oscillations) occurring during the recording period and these had a duration of 88.6±10.1 min per oscillation (range: 59.7–169.3 min), and an amplitude of 4.5±1.2% (range: 2.2–6.1%). Average total duration of oscillatory activity in these mice was 7.1±0.5 hr (range: 6.1–8.0 hr); significantly shorter

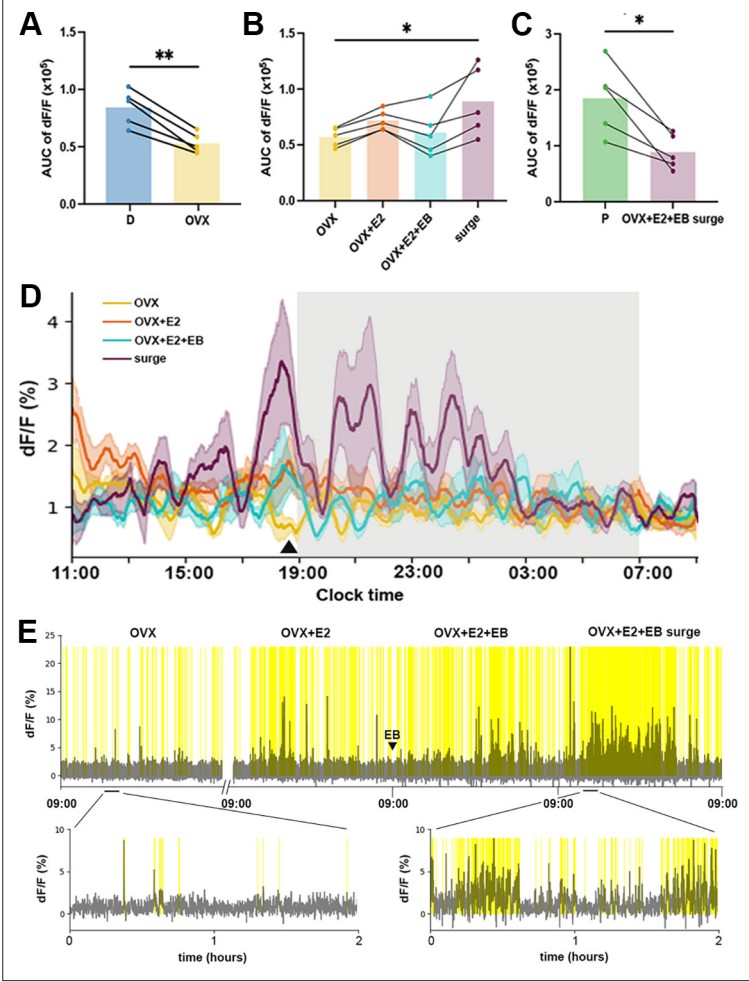

**Figure 9.** Changing rostral periventricular area of the third ventricle kisspeptin (RP3V^KISS) neuron baseline and transient activity following OVX+ estrogen treatment regimens. (**A**) Mean area under the curve (AUC) of 22 hr recordings for mice in diestrus (D) and following their ovariectomy (OVX), paired t-test: *p*=0.002 (**B**) Mean AUC calculated in OVX, OVX+E2, OVX+E2+EB, and OVX+E2+EB surge conditions. Friedman test: $\chi^2$=10.68, *p*=0.0055, Dunn's multiple comparisons test *p*=0.021. (**C**) Mean AUC from intact proestrous mice (**P**) compared to the same mice under OVX+E2+EB surge conditions, paired t-test: *p*=0.014. Each dot represents one animal (N=5). (**D**) Mean moving average of baseline recordings over 22 hr recordings showing variations after OVX (yellow), five days after estradiol implant (OVX+E2, orange), on the day of EB injection (OVX+E2+EB, aqua), and on the day of OVX+E2+EB surge (plum). Shaded regions around the traces indicate SEMs of corresponding colors (n=5). Triangle marks the pre-lights-off oscillation. Light-off period is represented by gray shaded area. (**E**) Example of high frequency transient activity from a representative mouse 7 days after OVX and then continuously for three days encompassing the fifth day after E2 implant (OVX+E2), on the day of EB injection (OVX+E2+EB), and the day of the expected OVX+E2+EB luteinizing hormone (LH) surge. Triangle indicates time of estradiol benzoate (EB) injection. Identified significant transients are highlighted in yellow. Expanded views of the traces are shown below.

compared to that of proestrus (12.7±0.7 hr; unpaired t-test, *p*=0.001). During the other treatment conditions, one out of five females exhibited a single identified oscillation on the day of EB injection (OVX+E2+EB).

Notably, the oscillation observed ~1 hr before lights-off in intact mice disappeared after OVX but appeared to be restored under all estrogen replacement conditions (***Figure 9D***).

The high frequency transients (***Figure 9E***) were evaluated across all four conditions. The frequency of transients was significantly elevated during the OVX+E2+EB surge condition compared to OVX (Friedman test: $\chi^2$=14.04, *p*<0.0001; Dunn's multiple comparisons tests: *p*=0.0014) (***Figure 8E***, ***Table 1***). The durations of transients were not different across conditions (***Table 1***, Repeated measures one-way ANOVA: *F*=0.50, *p*=0.53). Although repeated measures one-way ANOVA revealed

**Table 1.** High frequency transients parameters in different treatment groups (Mean ± SEM).

| Treatment | Frequency (Hz) | Duration (seconds) | Amplitude (dF/F %) |
|---|---|---|---|
| OVX | 0.0020±0.00046 | 14.36±1.85 | 2.30±0.22 |
| OVX+E2 | 0.0038±0.00028 | 14.34±0.86 | 2.47±0.26 |
| OVX+E2+EB | 0.0044±0.00029 | 13.40±0.79 | 2.55±0.27 |
| OVX+E2+EB surge | 0.0120±0.00190 | 13.13±1.00 | 3.03±0.39 |

a significant main effect of estrogen treatment on transient amplitude ($F$=8.1, $p$=0.039), post-hoc analysis using Tukey's multiple comparisons tests did not detect any significant pairwise differences between individual conditions (*Table 1*).

## Discussion

We report here that RP3V[KISS] neurons exhibit a marked increase in population activity on the afternoon of proestrus that consists of ~90 min duration oscillations, associated with high frequency transient activity, that last for approximately 13 hr. The first half of this activation period almost certainly drives the proestrous LH surge (*Goodman et al., 2022*; *Kauffman, 2022*) while the role of the latter activity remains less certain. We show that this behavior of RP3V[KISS] neurons is driven by circulating estradiol, although the standard OVX+E2+EB model of estrogen replacement used here appears insufficient to drive the normal magnitude of RP3V[KISS] neuron activity.

The activity of the RP3V[KISS] neuron population is relatively stable during metestrus and diestrus with only very occasional excursions from basal activity. This changes markedly on the afternoon of proestrus when substantial baseline oscillations begin and are recorded for the next ~13 hr. While this change in RP3V kisspeptin neuron activity almost certainly drives the LH surge, the precise temporal relationship between the initial onset of RP3V[KISS] neuron activity and LH secretion remains unclear. The resolution of blood sampling (hours) was poor compared to that of the neural recordings (seconds),

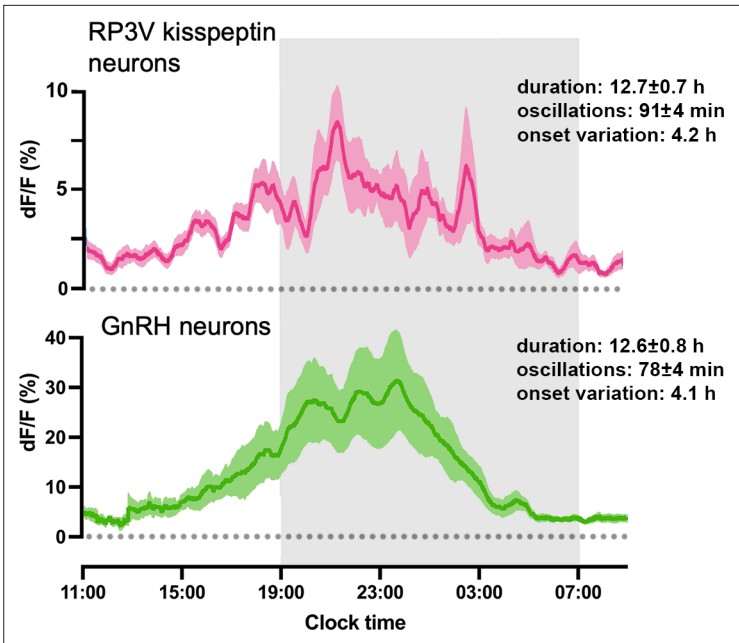

**Figure 10.** Comparison between rostral periventricular area of the third ventricle kisspeptin (RP3V[KISS]) and GnRH neuron activity patterns on proestrus. Mean 30 min moving average of 22 hr GCaMP recordings across proestrus from RP3V[KISS] neurons (pink; taken from *Figure 5F*; N=8) and GnRH neuron dendron activity (green; raw data obtained from *Han et al., 2025*; n=7) processed in the same manner. Shaded regions around the traces indicate ± SEMs of corresponding colors. Text provides mean ± SEM duration of total elevated activity, duration of each oscillation, and variation in onset of oscillations between animals.

and insufficient to establish any precise correlation. The abrupt activation of GnRH neurons by exogenous kisspeptin (*Han et al., 2005*) or LH secretion by optogenetic activation of RP3V[KISS] neurons (*Piet et al., 2018*) indicates that LH secretion would likely be elevated soon after any increment in kisspeptin transmission. Indeed, the in vivo temporal profiles of RP3V kisspeptin neurons and GnRH neuron dendrons are near identical across proestrus suggesting a high degree of temporal fidelity (*Figure 10*). However, more detailed LH sampling protocols will need to be implemented alongside GCaMP fibre photometry to examine the precise relationship between the start of RP3V kisspeptin neuron activity and LH secretion.

We find that RP3V[KISS] neuron activity continues for many hours past the peak of the LH surge. This is the same as the profile of GnRH neuron activity in the mouse on proestrus (*Han et al., 2025*; *Liu et al., 2025*; *Figure 10*) and long-standing observations that GnRH secretion at the median eminence greatly outlast the LH surge in multiple species (*Sarkar et al., 1976*; *Pau et al., 1993*; *Karsch et al., 1997*). While it remains unclear what the role of GnRH secretion after the LH surge may be (*Skinner and Caraty, 2002*), it is very likely that the extended period of GnRH and LH secretion is driven by prolonged RP3V[KISS] neuron input to the GnRH neurons.

The impact of long-duration RP3V[KISS] neuron activity on other neural networks in the forebrain during proestrus remain unknown. The RP3V[KISS] neurons project widely throughout the limbic forebrain (*Yeo and Herbison, 2011*) and may conceivably operate to coordinate multiple functions relevant to reproduction at proestrus. Indeed, commensurate with the long period of elevated RP3V[KISS] neuron activity detected here following the surge, these neurons have also been proposed to modulate female mate preference and copulatory behavior (*Hellier et al., 2018*) that would typically occur a few hours after the surge. At present, it remains unknown whether the sub-population of RP3V[KISS] neurons innervating GnRH neurons also send collaterals to other brain regions.

The recent ability to record the activity of the GnRH neuron population in vivo revealed that their elevated activity on the afternoon of proestrus occurred in an oscillatory manner with a period of ~78 min (*Han et al., 2025*). This provides a striking parallel with the ~91 min-duration oscillations now detected in RP3V[KISS] neurons activity over the exact same time period. This suggests that the unusual oscillatory pattern of GnRH neuron activity on proestrus is being driven by RP3V[KISS] neurons. Indeed, the entire profile of RP3V[KISS] and GnRH neuron activity across proestrus is strikingly similar (*Figure 10*) making it very likely that GnRH neuron activity on proestrus is patterned almost entirely by the RP3V[KISS] neurons. Nevertheless, a small role may also exist for the ARN kisspeptin neuron pulse generator as it remains active during the ascending phase of the proestrous LH surge (*McQuillan et al., 2019*; *Han et al., 2025*). As such, the episodic release of kisspeptin from the ARN pulse generator on GnRH neuron dendrons may potentiate early GnRH secretion driven by RP3V[KISS] neurons. However, the functional impact of ARN kisspeptin neurons on the surge remains unclear in mice. The deletion of ARN kisspeptin neurons was not found to have any effect on the LH surge (*Coutinho et al., 2024*) while selective removal of kisspeptin from ARN neurons resulted in a decrease in the amplitude of the LH surge (*Velasco et al., 2023*).

It is interesting to speculate on how oscillatory activity may be generated in RP3V[KISS] neurons and what role it may play. We find that the baseline GCaMP oscillations are associated with intense high frequency transient activity with each transient having a duration of approximately 10 s. This may represent the signature of single or small groups of RP3V[KISS] neurons exhibiting 10 s burst firing that then go on to synchronise into 90 min episodes of activity. In acute brain slices, RP3V[KISS] neurons often exhibit spontaneous burst firing (*Jamieson and Piet, 2022*) and each burst is known to generate 5–10 s-duration calcium transients in GCaMP recordings (*Piet et al., 2015*). It is also interesting to note that the optogenetic induction of burst firing in RP3V[KISS] neurons is the most efficient mode for activating GnRH neurons and LH secretion (*Piet et al., 2018*). How RP3V[KISS] neurons may synchronise their activity is unknown. It is possible that, like ARN kisspeptin neurons, they use direct recurrent collateral innervation to facilitate periods of synchronous bursting. Alternatively, synchronicity may be generated within a wider network of RP3V circuitry and/or by afferent inputs. The physiological roles and necessity for episodic activity in GnRH neurons, driven by RP3V[KISS] neurons, are also unknown. As these neurons are intensely active for prolonged periods of over 12 hr, it is possible that recurrent episodes of inhibition protect RP3V[KISS] and GnRH neurons from excitotoxicity.

It is well established that the onset of the LH surge is entrained by circadian inputs in rodents (*Tonsfeldt et al., 2022*; *Piet, 2023*). However, this entrainment is not highly time-locked compared to

other circadian-timed events, such as locomotion (*Starnes and Jones, 2023*). We show here that the onset of RP3V[KISS] neuron activity between mice, and even within the same mouse, varies by several hours around the time of 'lights off' on proestrus. The exact same observations have been made for GnRH neuron activation and the LH surge on proestrus (*Miller et al., 2004*; *Minabe et al., 2011*; *Czieselsky et al., 2016*; *Yeo et al., 2025*). The mechanisms responsible for this variability are not known. Although only three of the OVX+E2+EB mice exhibited RP3V[KISS] neuron oscillations, they also initiated at different times suggesting perhaps that the variability is not due to differences in the time estrogen peaks.

Given this variability, we were surprised to observe a tightly time-locked RP3V[KISS] neuron increment in activity almost exactly 1 hr before 'lights off' on proestrus, as well as on every other day of the cycle. While this would appear to be a more rigid circadian input to the RP3V[KISS] neurons, its role in triggering the LH surge is unclear and will require further investigation. Interestingly, OVX mice were the only group in which we did not observe this circadian event. This dependence on estrogen is reminiscent of the excitatory effect of vasopressin on RP3V[KISS] neuron firing that is unchanging in different estrogenic states but absent in OVX mice (*Piet et al., 2015*).

We find that the activity of the RP3V[KISS] neuron population is reduced 7 days following ovariectomy but that relatively little change is evoked by a sequential regimen of estrogen replacement until the day of the expected surge when proestrus-like oscillatory activity emerges. This is consistent with the concept of the surge being generated by rising estrogen levels that harness slow transcriptional mechanisms to alter RP3V[KISS] neuron excitability (*Glidewell-Kenney et al., 2007*; *Herbison, 2015*). Increments in circulating estradiol either in the OVX+E model or during diestrus, when estradiol levels peak (*Wall et al., 2023*), appear to have little immediate effect on RP3V[KISS] neuron activity patterns. This is broadly in agreement with acute brain slice studies that have found reduced RP3V[KISS] neuron firing rates following OVX but then rather few consistent changes in firing across the estrous cycle itself (*Jamieson and Piet, 2022*). Nevertheless, it is clear that estrogen treatment of OVX mice slowly up-regulates sodium ($I_{NaP}$), calcium ($I_T$), and hyperpolarization-activated ($I_h$) ion channels in RP3V[KISS] neurons, that would all be expected to result in increased excitability (*Jamieson and Piet, 2022*). Once established, these changes in ion channel expression are presumably at least partly responsible for the enhanced oscillatory-like activity patterns of RP3V[KISS] neurons held in check until the day of the surge.

While the same pattern of RP3V[KISS] neuron activity is observed on the afternoon of proestrus and in the OVX+E2+EB model, it is evident that the magnitude of change is dramatically reduced in the latter. We did not assess surge LH levels in the OVX+E2+EB mice so it is possible that the two animals that did not exhibit oscillatory activity did not surge. For those that did, it is likely that they exhibited LH surges with a lower amplitude compared to proestrus, as prior work from the laboratory using the same model has shown that peak LH surge levels in OVX+E2+EB mice (7.7±0.7 ng/mL) are half of that observed in proestrus mice (14.5±1.5 ng/mL) (*Czieselsky et al., 2016*). This inability to recreate full proestrous-like LH surge levels is typical of estrogen-replaced OVX mouse models (*Bronson and Vom Saal, 1979*; *Dror et al., 2013*; *Silveira et al., 2017*). We would suggest that the reduced amplitude of the LH surge in estrogen-replaced OVX models likely originates from sub-optimal activation of the RP3V[KISS] neurons. This may be due to a lack of sufficient progesterone or ovarian factors in estrogen-replaced OVX models (*Micevych and Sinchak, 2011*). However, it is important to note that the seminal experiments of *Bronson and Vom Saal, 1979* identified that a full LH surge can be achieved in OVX+E2+EB mice with careful attention to the timing of the EB injection. Our present OVX+E2+EB protocol matches the most efficacious protocol of Bronson and Vom Saal with the exceptions of using 4 compared to 5 µg E2 capsules and a 12:12 rather than 14:10 lighting regimen.

In summary, we demonstrate that RP3V[KISS] neurons exhibit an unusual, extended period of baseline oscillatory activity on proestrus. The genesis of oscillatory RP3V[KISS] neuronal activity is critically dependent on long-term exposure to estradiol with a likely involvement of circadian inputs. This oscillatory RP3V[KISS] neuron activity almost certainly drives the same pattern of GnRH neuron activity critical for triggering the preovulatory LH surge. It is possible that oscillatory RP3V[KISS] neuron activity existing beyond the generation of the LH surge operates to coordinate other neural circuits controlling reproduction, such as those underlying female sexual behavior (*Hellier et al., 2018*).

## Materials and methods

### Animals

Adult *B6(129S4)-Kiss1^{tm1.1(cre/EGFP)Rpa}/J* (Palmiter *Kiss1^{Cre/+}* (v2), JAX stock #033169) (*Padilla et al., 2018*) female mice were group-housed in conventional cages with environmental enrichment under conditions of controlled temperature (22±2°C) and lighting (12 hr light/12 hr dark cycle; lights on at 07:00) with ad libitum access to water and low-phytoestrogen food (Tekland 2016, Envigo RMS, UK) and water. Following surgery, mice were single-housed in conventional cages under the same conditions until the end of the study. All animal experimental protocols were approved by the University of Cambridge Animal Welfare and Ethics Review Body under UK Home Office licenses P174441DE and PP9818192.

### Stereotaxic implantation of optic fibres and AAV injections

Adult female mice (10–14 weeks-old) were anaesthetized with 2% isoflurane and placed in a stereotaxic frame with buprenorphine (0.05 mg/kg, s.c.) and meloxicam (5 mg/kg, s.c.) analgesia. Dexamethasone (10 mg/kg, s.c.) was used to prevent cranial swelling. A custom-built Hamilton syringe apparatus, holding a 5 μL Hamilton syringe (Hamilton, catalogue No. 7634–01) with a 29-gauge bevelled needle (Hamilton, catalogue No. 90029), was filled with 2.0 μL of recombinant Cre-dependent AAVs encoding GCaMP6s (AAV9CAG.FLEX.GCaMP6s.WPRE.SV40, Addgene 100842-AAV9, titre: 2×10^{13} genome copy/mL). Mice for photometry experiments received a single unilateral injection of 2.0 μL AAV at coordinates 0.8 mm anterior to bregma, 0.25 mm lateral to the superior sagittal sinus, and 5.0 mm deep. Mice for immunohistochemistry characterization received bilateral AAV injections. After a unilateral viral injection, an indwelling tapered optical fibre (400 μm diameter, 0.66 NA, 3.5 mm taper length, Doric Lenses, Quebec, Canada) was implanted directly alongside the RP3V on the same side (0.55 mm anterior to bregma, 0.22 mm lateral to the superior sagittal sinus, 5.2 mm deep). One week following surgery, all animals were handled daily and habituated to a photometry recording set-up for at least three weeks. Estrous cycle stage was assessed by vaginal lavage each morning.

### Estrogen-induced surge model

All estrogen-induced surge studies in the laboratory use the OVX+E2+EB protocol of Bronson and Vom Saal with slight modifications (*Bronson and Vom Saal, 1979*). One week after OVX, mice were implanted subcutaneously with silastic implants (id, 1.02 mm; od, 2.16 mm, Dow Corning Corp) containing 4 μg 17β-estradiol (E2, Sigma-Aldrich, catalogue No. E8875) per 20 g body weight. This dose of 4 μg has been shown to return E2 levels to diestrous concentrations (*Porteous et al., 2021*). Six days after capsule implantation, mice were given a subcutaneous injection of 1 μg β-estradiol 3-benzoate (EB, Sigma-Aldrich, catalogue No. E8875) in 100 μL sesame oil at 09:00 hr, 2 hr after lights on. This protocol evokes an LH surge the following day (*Figure 7*).

To assess the relationship between RP3V^{KISS} neuron and the proestrous LH surge, female mice were attached to the fibre photometry system in the morning of proestrus and tail-tip blood samples (5 μL) collected every 2–4 hr over a period of 8–10 hr (*Czieselsky et al., 2016*). Levels of LH were measured by LH ELISA (*Kreisman et al., 2022*) with a sensitivity of 0.07 ng/mL and intra-assay and inter-assay coefficients of variation of 8.2% and 11.1%, respectively.

### Immunohistochemistry

Female mice receiving bilateral injections of Cre-dependent AAVs encoding GCaMP6s were perfused with ice-cold 4% paraformaldehyde (PFA) on the afternoon of proestrus at least three weeks after viral injections. Brains were removed and post-fixed in 4% PFA for 4 hr before transferred to 30% sucrose. Forty-μm-thick coronal sections were cut on a freezing microtome for immunofluorescence. Brain sections were incubated with polyclonal rabbit anti-kisspeptin antisera, raised against the final ten amino acids of murine kisspeptin (1:5,000; AC566, Dr. Alain Caraty, Nouzilly, France) and chicken anti-GFP (1:5,000 Aves Lab, catalogue No. GFP-1020) for 72 hr. Brain sections were then washed and incubated with biotinylated goat anti-rabbit IgG (1:500; Vector Laboratories, catalogue No. BA-1000), followed by Streptavidin Alexa Fluor 568 (1:500; Invitrogen, Thermo Fisher, catalogue No. S11226) and goat anti-chicken Alexa Fluor 488 (1:1000; Molecular Probes, Thermo Fisher, catalogue No. A-11039).

Immunofluorescent images were taken on a laser scanning confocal microscope (Zeiss LSM900) using a 20 x objective lens in z-stacks with 2 μm steps across the depth of the slide. The number of

GFP-expressing, kisspeptin-expressing, and co-expressing cell bodies were manually counted using ImageJ.

## GCaMP6 fibre photometry and blood sampling

GCaMP6 fibre photometry was set up as previously described (*Clarkson et al., 2017*). Fluorescence signals were sampled at 10 Hz using a scheduled mode (2 s on/ 2 s off) of light emission with Tussock Innovation software available from Argotech (https://www.argotech.co.nz/fibre-photometry). Recordings were made for 22 hr from mice across each stage of the estrous cycle, 1 week after ovariectomy, and during different times of the OVX+E2+EB surge protocol (*Figure 7*); with all starting 4 hr after lights-on.

Analysis of fluorescent signals was performed in MATLAB with the subtraction of the 405 signal from the 465 signal to extract the calcium-dependent fluorescence signal. Fluorescent signals were then down-sampled by averaging each 2 s recording during the 4 s on-off period into a single data point. An exponential fit algorithm was used to correct for baseline shifts using a 6 hr window before the signal was calculated in dF/F (%) with the equation $dF/F = (F_{fluorescence} - F_{baseline})/ (F_{baseline}) \times 100$.

Numpy.trapezoid function was used in Python to calculate the definite integral of the curve, approximating the area under the curve (AUC) using the trapezoidal rule. Deconvolution of signals was achieved by applying the movmean algorithm in MATLAB to extract a 30 min rolling average (*Han et al., 2025*). With this approach, slow oscillations in baseline were then identified by findpeaks function in MATLAB with (a) a minimum peak height set as two standard deviations above the mean calcium signal from 11:00-13:00 hr, (b) minimum peak distance as 30 min, and (c) minimum peak prominence as 0.15 of maximum signal in proestrous recording for each animal. Time taken between one trough to another in individual oscillations was calculated as the duration of each oscillation. The total duration of the surge signals was measured from their onset to offset. Onset was defined as the trough at the start of the first oscillation, and offset was defined as the trough at the end of the last oscillation.

To analyze the high frequency transients, the 30 min moving average was subtracted from the original signal, and an additional exponential fit algorithm with a 400 s window was applied in MATLAB to correct baseline shifts. To identify significant calcium transients, the method of Dombeck and colleagues (*Dombeck et al., 2010*) was adapted for GCaMP6s dynamics. First, the dF/F trace was z-scored such that its median was normalized to zero, and the standard deviation was set to one. Then, a series of thresholds iterated over the range 1–4 with step size 0.2: k = [1, 1.2, 1.4, …, 4] were defined. For each threshold k: positive and negative-going transients ($T_{pos}$ and $T_{neg}$) were identified. The z-scored trace was then thresholded such that z-score >k to generate a binary array. The segments of the array labeled as 1 (active transient) were classified as $T_{pos}$. The z-scored trace was also thresholded so that z-score < -k, producing another binary array, with segments labeled as 1 classified as $T_{neg}$. For each transient in $T_{pos}$, the transient significance was also assessed for its duration size s. The number of transients with sizes larger than s in both $T_{pos}$ and $T_{neg}$ were counted, as $n_{pos}$ and $n_{neg}$, respectively. If $n_{neg}/n_{pos} < 0.05$, the positive transients with sizes greater than s were identified as significant. Any part of the trace identified as significant under any combination of k and s was considered a significant transient.

## Statistical analysis

All statistical analyses were performed using GraphPad Prism 10 software. Each dataset was tested for normality with the Shapiro-Wilk test and visual inspection of Q-Q plots. Homogeneity of variance was tested by the F-test when comparing two groups or Bartlett's test when comparing three or more groups. If data are normally distributed and variances are homogeneous, one-way ANOVA, repeated measures one-way ANOVA with Geisser-Greenhouse correction, followed by Tukey's multiple comparisons test, and paired t-tests were applied. Non-parametric tests, such as the Friedman test followed by Dunn's multiple comparison tests were used when data are not normally distributed, or when the variances are not homogeneous. The threshold level for statistical significance was set at $p < 0.05$.

The ARRIVE guidelines were followed. All data represent biological replicates. As an observational study, randomization and blinding were not required. Sample sizes were determined based upon prior studies examining kisspeptin neurons with fibre photometry. There were no inclusion/exclusion criteria.

## Acknowledgements

This work was supported by the Wellcome Trust (212242/Z/18/Z) and a UKRI Medical Research Council Equipment Grant (MC-PC-MR-X012271/1). ZZ was supported by the UKRI Medical Research Council (MR N013433-1) and the Harding Distinguished Postgraduate Scholars Programme Leverage Scheme.

## Additional information

### Funding

| Funder | Grant reference number | Author |
|---|---|---|
| Wellcome Trust | 10.35802/212242 | Allan Edward Herbison |
| Medical Research Council | N013433-1 | Ziyue Zhou |
| UKRI Medical Research Council Equipment | MC-PC-MR-X012271/1 | Allan Edward Herbison |
| Harding Distinguished Postgraduate Scholars Programme | Leverage Scheme | Ziyue Zhou |

The funders had no role in study design, data collection and interpretation, or the decision to submit the work for publication. For the purpose of Open Access, the authors have applied a CC BY public copyright license to any Author Accepted Manuscript version arising from this submission.

### Author contributions

Ziyue Zhou, Formal analysis, Investigation, Methodology, Writing – original draft, Writing – review and editing; Cheng-Yu Huang, Formal analysis; Allan Edward Herbison, Conceptualisation, Funding acquisition, Writing – original draft, Project administration, Writing – review and editing

### Author ORCIDs

Cheng-Yu Huang ![ORCID] https://orcid.org/0000-0002-2153-4584
Allan Edward Herbison ![ORCID] https://orcid.org/0000-0002-9615-3022

### Ethics

All animal experimental protocols were approved by the University of Cambridge Animal Welfare and Ethics Review Body under UK Home Office licenses P174441DE and PP9818192.

Joint Public Review: https://doi.org/10.7554/eLife.109215.3.sa1
Author response https://doi.org/10.7554/eLife.109215.3.sa2

## Additional files

### Supplementary files

MDAR checklist

### Data availability

All data generated or analysed during this study are included in the manuscript and supporting files; source data files have been provided for all figures at the University of Cambridge Apollo Repository https://doi.org/10.17863/CAM.129370.

The following dataset was generated:

| Author(s) | Year | Dataset title | Dataset URL | Database and Identifier |
|---|---|---|---|---|
| Zhou Z, Huang C-Y, Herbison AE | 2026 | Data supporting "Prolonged oscillating preoptic area kisspeptin neuron activity underlies the preovulatory luteinizing hormone surge in mice" | https://doi.org/10.17863/CAM.129370 | Cambridge Apollo Repository, 10.17863/CAM.129370 |

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
