## [Editor Report · eLife Assessment]

This **fundamental** work advances our understanding of the role of kisspeptin neurons in regulating the luteinizing hormone (LH) surge in females. The study uses cutting-edge techniques to provide **compelling** and rigorous data supporting a critical role of RP3V kisspeptin neurons in the neuroendocrine LH surge process. This research will be of interest to reproductive biologists and neuroscientists studying the female ovarian cycle. Continuing to examine the complexities of the LH surge and the neuronal populations involved, as done in this study, is critical for developing therapeutic treatments for women's reproductive disorders.

---

## [Referee Report · Joint Public Review]

Summary:

This is an excellent, timely study investigating and characterizing the underlying neural activity that generates the neuroendocrine GnRH and LH surges that are responsible for triggering ovulation. Abundant evidence accumulated over the past 20 years implicated the population of kisspeptin neurons in the hypothalamic RP3V region (also referred to as the POA or AVPV/PeN kisspeptin neurons) as being involved in driving the GnRH surge in response to elevated estradiol (E2), also known as the estrogen positive feedback. However, while former studies used cfos coexpression as a marker of RP3V kisspeptin neuron activation at specific times and found that this correlates with the timing of the LH surge, detailed examination of the live in vivo activity of these neurons before, during, and after the LH surge, remained elusive due to technical challenges. In this exciting study, Zhou and colleagues use fiber photometry to measure the long-term synchronous activity of RP3V kisspeptin neurons across different stages of the mouse estrous cycle, including on proestrus when the LH surge occurs, as well as in a well-established OVX+E2 mouse model of the LH surge. For this they used kiss-Cre female mice that were injected with a Cre-dependent AAV injection containing GCaMP6, in order to measure the neuronal activation of RP3V Kiss1 cells.

The authors report that RP3V kisspeptin neuronal activity is low on estrous and diestrus, but increases on proestrus several hours before the late afternoon LH surge, mirroring prior reports of rising GnRH neuron activity in proestrus female mice. The measured increase in RP3V kisspeptin activation is long, spanning ~13 hours in proestrus females and extending well beyond the end of the LH secretion, and is shown by the authors to be E2 dependent. In addition, an intriguing cyclical oscillation in kisspeptin neural activity every 90 minutes exists, which may offer critical insight into how the RP3V kisspeptin system operates.

The compelling methodology allowed the authors to measure RP3V neuronal activation across multiple ovarian cycles in the same mouse, which demonstrated that the timing of the LH surge is variable across cycles, even within the same mouse. In addition, the authors demonstrated using the same females, that ovariectomy resulted in very little neuronal activity in RP3V kisspeptin neurons. When these ovariectomized females were treated with estradiol benzoate (EB) and an LH surge was induced, there was an increase in RP3V kisspeptin neuronal activation, as was seen during proestrus. However, the magnitude of the change in activity was greater during proestrus than during the EB-induced LH surge. Interestingly, the authors noted a consistent peak in activity about 90 minutes prior to lights out on each day of the ovarian cycle and during EB treatment, but not in ovariectomized females. The functional significance of this consistent neuronal activity at this time remains to be determined. In summary, the data from these experiments is compelling and supports the hypothesis in the field that the RP3V kisspeptin neurons regulate the LH surge.

Strengths:

- The study is well designed, uses proper controls and analyses, has robust data, and the paper is nicely organized and written.

- The study is well done and complete, looking at neuronal activation at each stage of the ovarian cycle and then additionally, how neuronal activation in ovariectomized and ovariectomized + EB females compares to that of gonad-intact females. Though not part of this study, the comparison of neuronal activation of GnRH neurons during the LH surge to the current data was convincing, demonstrating a similar pattern of increased activation that precedes the LH surge.

- The authors provide a technical advance for the field in the ability to accurately measure RP3V kisspeptin neuron activity in actively awake, live mice for long periods of time, spanning different cycle stages. This approach offers novel and useful insights into the impact of E2 and circadian cues on the electrical activity of RP3V kisspeptin neurons.

- The within-subjects design used in these experiments is a major strength because it allowed the authors to collect data across multiple ovarian cycles, following ovariectomy, and then with EB treatment. The variability in neuronal activity surrounding the LH surge across ovarian cycles in the same animals is interesting and could not be achieved without this within-subjects design.

- The inclusion and comparison of ovary-intact females and OVX+E2 female is valuable to help test mechanisms under these two valuable LH surge conditions, and allows for further future studies to tease apart minor differences in the LH surge pattern between these 2 conditions.

- The discovery of cyclical oscillation in RP3V kisspeptin neural activity every 90 minutes is intriguing and interesting, and may offer critical insight into how the RP3V kisspeptin system operates, which can be further tested in future studies.

Weaknesses:

- LH levels were not measured in many mice or in robust temporal detail, to allow a more detailed comparison between the fine-scale timing of RP3V neuron activation with onset and timing of LH surge dynamics. While the "peak LH" occurred 3.5 hours after the first RP3V kisspeptin neuron oscillation, it is likely that LH values start to increase several hours before the peak LH, closer to when the first RP3V kisspeptin neuron activity first occurs. Therefore, the onset of the LH surge is likely to be closer to the beginning of the RP3V kisspeptin activity, but future studies are needed to study this timing.

- One minor concern is that LH levels were not measured in the ovariectomized females during the expected time of the LH surge. The authors suggest that the lower magnitude of activation during the LH surge in these females, in comparison to proestrus females, may be the result of lower LH levels. It's hard to interpret the difference in magnitude of neuronal activation between EB-treated and proestrus females without knowing LH levels. In addition, it's possible that an LH surge did not occur in all EB-treated females, and thus, having LH levels would confirm the success of the EB treatment.

- The authors nicely show that there is some variation (~2 hours) in the peak of the first oscillation in cycling proestrus females. By contrast, the small sample size for OVX+E2 females did not permit a similar rigorous analysis of temporal variability under such estrogen-controlled conditions, which will need to be studied in future projects.

Comments on revisions:

The authors have revised the manuscript adequately. There are no further recommended edits or revisions.

---

## [Author Response]

The following is the authors’ response to the original reviews.

**Joint Public Review:**
Weaknesses:(1) LH levels were not measured in many mice or in robust temporal detail, such as every 30 or 60 min, to allow a more detailed comparison between the fine-scale timing of RP3V neuron activation with onset and timing of LH surge dynamics.

Please see “Recommendations for Authors” below.

(2) The authors report that the peak LH value occurred 3.5 hours after the first RP3V kisspeptin neuron oscillation. However, it is likely, and indeed evident from the 2 example LH patterns shown in Figures 3A-B, that LH values start to increase several hours before the peak LH. This earlier rise in LH levels ("onset" of the surge) occurs much closer in time to the first RP3V kisspeptin neuron oscillatory activation, and as such, the ensuing LH secretion may not be as delayed as the authors suggest.

Please see “Recommendations for Authors” below.

(3) The authors nicely show that there is some variation (~2 hours) in the peak of the first oscillation in proestrus females. Was this same variability present in OVX+E2 females, or was the variability smaller or absent in OVX+E2 versus proestrus? It is possible that the variability in proestrus mice is due to variability in the timing and magnitude of rising E2 levels, which would, in theory, be more tightly controlled and similar among mice in the OVX+E2 model. If so, the OVX+E2 mice may have less variability between mice for the onset of RP3V kisspeptin activity.

Please see “Recommendations for Authors” below.

(4) One concern regarding this study is the lack of data showing the specificity of the AAV and the GCaMP6s signals. There are no data showing that GCaMP6s is limited to the RP3V and is not expressed in other Kiss1 populations in the brain. Given that 2ul of the AAV was injected, which seems like a lot considering it was close to the ventricle, it is important to show that the signal and measured activity are specific to the RP3V region. Though the authors discuss potential reasons for the low co-expression of GCaMP6 and kisspeptin immunoreactivity, it does raise some concern regarding the interpretation of these results. The low co-expression makes it difficult to confirm the Kiss1 cell-specificity of the Cre-dependent AAV injections. In addition, if GFP (GCaMP6s) and kisspeptin protein co-localization is low, it is possible that the activation of these neurons does not coincide with changes in kisspeptin or that these neurons are even expressing Kiss1 or kisspeptin at the time of activation. It is important to remember that the study measures activation of the kisspeptin neuron, and it does not reveal anything specific about the activity of the kisspeptin protein.

Please see “Recommendations for Authors” below.

(5) One additional minor concern is that LH levels were not measured in the ovariectomized females during the expected time of the LH surge. The authors suggest that the lower magnitude of activation during the LH surge in these females, in comparison to proestrus females, may be the result of lower LH levels. It's hard to interpret the difference in magnitude of neuronal activation between EB-treated and proestrus females without knowing LH levels. In addition, it's possible that an LH surge did not occur in all EB-treated females, and thus, having LH levels would confirm the success of the EB treatment.

Please see “Recommendations for Authors” below.

(6) This kisspeptin neuron peak activity is abolished in ovariectomized mice, and estradiol replacement restored this activity, but only partially. Circulating levels of estradiol were not measured in these different setups, but the authors hypothesize that the lack of full restoration may be due to the absence of other ovarian signals, possibly progesterone.

Please see “Recommendations for Authors” below.

(7) Recordings in several mice show inter- and intra-variability in the time of peak onset. It is not shown whether this variability is associated with a similar variability in the timing of the LH surge onset in the recorded mice. The authors hypothesized that this variability indicates a poor involvement of the circadian input. However, no experiments were done to investigate the role of the (vasopressinergic-driven) circadian input on the kisspeptin neuron activation at the light/dark transition. Thus, we suggest that the authors be more tentative about this hypothesis.

Please see “Recommendations for Authors” below.

**Recommendations for the authors:**
(1) The study measured LH levels over time in just 5 female mice, a small sample size given the variability between mice. Having said that, n=5 is an OK starting point but the LH values are only shown for 2 mice, and there are no graphs or presentation of mean LH levels over time for all 5 mice. Figure 3 would greatly benefit from graphing and statistical analyses of the LH levels for all 5 mice (mean line graphs over time or similar). The authors report the mean "peak LH" level in the text, but it would be important to show and graph all the LH values over time (either by clock time or time relative to start of first RP3V oscillation or both), to allow the reader to compare the LH pattern to the RP3V kisspeptin neuron activity over time.

We share the Reviewer’s frustration regarding the lack of detailed LH time points to correlate with the changes in GCaMP signal. Certainly, it was our intention to do better. However, with the benefit of actually being able to monitor surge progress through RP3V neuron activity in real time, we found that frequent blood sampling could often interfere with the normal dynamic of surge activity. One some occasions, the RP3V kisspeptin neuron oscillations would stop abruptly mid- or early-surge while on others it would stop and then start again. Knowing that this was not the normal profile, we resorted to taking as few blood samples as possible, trying primarily to get what we thought might be the “peak” LH surge level. We acknowledge that this is not ideal, and leaves open the important question around the precise relationship of the beginning of RP3V kisspeptin oscillations with LH secretion. Although not answering the question directly, this was part of the motivation for the last figure which emphasizes how the RP3V kisspeptin neuron activity and GnRH neuron dendron activity are essentially identical at the time of the surge. We have re-written the relevant section of the Discussion to be more circumspect.

(2) The authors report and discuss that the peak LH value occurred 3.5 hours after the first RP3V kisspeptin neuron oscillation but it is likely, and indeed evident from the 2 example LH patterns shown in Figs 3A-B, that LH values start to increase several hours earlier, well before the peak LH. Thus, the rise in LH levels during the surge starts much closer in time to the first RP3V kisspeptin neuron oscillatory activation, which the authors don't analyze. For example, the 2nd LH value for the 2 representative mice shown in Figure 3 is notably higher than the 1st LH value of those mice, even though the peak value has not yet been attained. Even with the LH levels only being measured here every couple hours, this "first detected rise in LH" be at least be graphed and/or analyzed relative to the timing of kisspeptin neuron activity, and commented on in the Discussion.

As above.

(3) It is unclear if the variation (~2 hours) in the peak of the first oscillation in proestrus females is the same as in OVX+E2 females, or was the variability smaller or absent in OVX+E2 females versus proestrus? The variability observed in proestrus mice is likely due to variability in timing and magnitude of rising E2 levels, which would may be more tightly controlled and similar among mice in the OVX+E2 model. If so, the OVX+E2 mice might display less variability for the timing of the RP3V kisspeptin activity "onset". This measure would be important to analyze here and to discuss, given that many labs around the world often use an OVX+E2 model.

This is an interesting point given the dogma surrounding the role of the SCN in initiating the surge. Three of the five OVX+E2 mice exhibited clearly discernible GCaMP oscillations that started at approximately noon, 1pm and 2pm. While this sample is very small, it does suggest that the onset of RP3V kisspeptin neuron activity is variable as found in proestrous mice. We have indicated this cautiously given the sample size.

(4) If looking at kisspeptin immunoreactivity is problematic, is it possible to look at Kiss1 RNA levels or to look at Cre-recombinase protein levels? While the Cre-recombinase would just be a proxy for Kiss1/kisspeptin, it may result in higher expression and better co-localization with the GCaMP6s.

Yes, RNAscope would likely be the ideal method to settle this long running issue of apparently poor Kiss-cre targeting in the RP3V. Unexpectedly, however, we found that the mCherry probe bound to Kiss1 in our attempts at an RNAscope evaluation. The use of Cre as a proxy for identifying kisspeptin neurons would almost certainly generate better co-localization as Cre is being used to target GCaMP.

Minor(1) It was not clear in the manuscript how many cells were counted or contributed to the neuronal activation data. Is it the entire population of RP3V Kiss1 cells? Just a subset? How much variability is there in the number of cells measured/counted between animals? Presumably, the brains were extracted to confirm the placement of the optic fiber. Were there neuroanatomical studies also done on these animals to confirm how many cells express GFP (GCaMP6) and the correct placement and specificity of the AAV? Is there any potential that cells in the BnST or even the ARC took up the virus and were included in these measurements?

It is very difficult if not impossible to establish just how many RP3V kisspeptin neurons contribute to the GCaMP population signal using fibre photometry. This will depend on levels of AAV transfection, distance from the optic fibre, and the numbers of RP3V kisspeptin neurons actually involved in the surge mechanism. Of note, C-Fos data suggest that only around one-third of RP3V kisspeptin neurons are activated at the time of the surge. All fibre placements were subsequently shown to be running alongside GCaMP-expressing AVPV/anterior periventricular nucleus cells (now noted), but the numbers of transfected cells were not quantified. As shown in Fig.4, the GCaMP signal was very similar across all mice suggesting little variation in the relationship between transduction, fibre placement and distance.

The RP3V region is approximately 4-5 mm from the ARN. We felt that the possibility that an AAV injection in the RP3V would spill over into the ARN was so remote that we did not assess GCaMP expression in ARN kisspeptin neurons. We have previously determined for the ARN that recordable GCaMP fluorescence only occurs if the optic fibre is within 0.5 mm from GCaMP-expressing neurons. Ultimately, proof that we are not recording from ARN kisspeptin neurons comes from the very different activity patterns reported here for RP3V neurons compared to the kisspeptin pulse generator. We did not see any GCaMP expressed in the BNST.

(2) If it is possible to measure LH levels in the EB surge animals, it would be helpful, at least to confirm that they did surge and to support the proposed idea that LH surge levels are lower in that model.

Unfortunately, as acknowledged in the original text we did not take blood samples from these mice so do not have the data. However, as noted, other studies undertaken by us using the same EB surge paradigm show that peak LH levels are much lower compared to proestrus. In retrospect we do agree that this would have been useful and particularly to establish whether each mouse did show a surge as two of the OVX+EB mice failed to show typical surge-associated oscillations. We have noted this in the Discussion.

(3) For Figure 4F, please add a gray shaded box to the graph to denote the "dark" period (lights off), as was done for Figures 2 and 3. This is important because Figure 4F is making the point that there is a consistent 90-minute oscillation event right before lights off, so it would be helpful to denote the period of lights off on the graph.

There was in fact a very light grey shade, but we have now added a grey bar to make the dark period clearer.

(4) The Title of the paper should include the brain region because this is specifically the RP3V (or preoptic area "POA") kisspeptin neurons that are studied, not other kisspeptin cell populations.

We have added “preoptic area” to clarify

(5) The graphs in Figure 3C-D are from different mice and address a different question than the graphs in Figure 3A-B. This was a bit confusing, and it is recommended that the LH + RP3V kisspeptin activity experiment (Figures 3A-B) be its own figure, and the graphs looking at the detailed oscillatory patterns in Figures 3C-D be their own figure, as the latter are addressing a different question and don't have any LH data.

We have split the figure as requested.

(6) The tiny font size of the X and Y axes of Figures 2 and 3 is very small and hard to read. Can this text please be increased in size a little? By comparison, the font size of the X and Y axes of Figure 4 is bigger and more legible.

Changed.

(7) In the methods for fiber photometry, there is a sentence saying "Twenty two-hour recordings were made..." This was confusing, as it read as if there were twenty 2-hour recordings, when in fact it was one 22-hour recording. The authors should reword or use "22-hour" in this sentence.

Changed.

(8) It's a bit hard to see the difference in color between proestrus 1 and proestrus 2 (both blues) in Figure 6, especially when they overlap. It might be helpful to select a different color for one of them.

Changed.

(9) Is the virus from Addgene or just the plasmid? Did Addgene insert the plasmid into the virus, or was that done elsewhere? For purposes of replication, it might be helpful to state the plasmid that was used and the virus that was used, and their origins (e.g., if made by Addgene or donated by another investigator). I was not able to find the virus based on the Addgene number in the manuscript and was getting plasmids with different Addgene #s.

Apologies, the numbering was incorrect. We have now amended to 100842-AAV9 that was packaged by Addgene.